# Effect of Various Fly Ash and Ground Granulated Blast Furnace Slag Content on Concrete Properties: Experiments and Modelling

**DOI:** 10.3390/ma15093016

**Published:** 2022-04-21

**Authors:** Zhiwei Qu, Zihao Liu, Ruizhe Si, Yingda Zhang

**Affiliations:** 1Centre for Infrastructure Engineering and Safety, School of Civil and Environmental Engineering, The University of New South Wales, Sydney, NSW 2052, Australia; zhiwei.qu1@student.unsw.edu.au; 2Architecture Course, Graduate School of Environmental Engineering, The University of Kitakyushu, 1-1 Hibikino Wakamatsu, Kitakyushu 808-0135, Fukuoka, Japan; b0dbb419@eng.kitakyu-u.ac.jp; 3Institute of Civil Engineering Materials, School of Civil Engineering, Southwest Jiaotong University, Chengdu 610031, China; ruizhesi@swjtu.edu.cn

**Keywords:** concrete, mechanical properties, fly ash, GGBFS, autogenous shrinkage, prediction model

## Abstract

Concrete is known as the most globally used construction material, but it releases a huge amount of greenhouse gases due to cement production. Recently, Supplementary Cementitious Materials (SCMs) such as fly ash and Ground Granulated Blast Furnace Slag (GGBFS) have been widely used in concrete to reduce the cement content. However, SCMs can alter the mechanical properties and time-dependent behaviors of concrete and the early age mechanical properties of concrete significantly affect the concrete cracking in the engineering field. Therefore, evaluation of the development of the mechanical properties of SCMs-based concrete is vital. In this paper, the time development of mechanical properties of concrete mixes with various fly ash and GGBFS was experimentally investigated. Four different cement replacement levels including 0%, 20%, 30%, and 40% by fly ash and GGBFS as well as ternary binders were considered. Compressive strength, splitting tensile strength, flexural strength, and elastic modulus of concrete were measured until 28 days. Three additional concrete mixes with ternary binders were also cast to investigate the early-age autogenous shrinkage development until 28 days. In addition, prediction models in existing standards were used and compared to experimental results. The comparison results showed that the prediction models overestimated the compressive strength but underestimated the splitting tensile strength development and autogenous shrinkage. As a result, a model capturing the effect of fly ash and GGBFS on the development of compressive and splitting tensile strength is proposed to improve the prediction accuracy for current standards and empirical models.

## 1. Introduction

Concrete is known as the most globally used construction material due to its excellent performance of compressive strength and workability. The tensile strength of concrete can also be improved using rebar and fibres [1]. Due to the rapid development of urbanisation, the consumption of concrete has increased significantly, since it is used as the primary construction material. In addition, Ordinary Portland Cement (OPC) is utilised as the primary cementitious material in concrete. From 1880 to 1990, it was reported that the global consumption of OPC has increased to 1.3 billion tons. Annual global cement consumption is expected to increase up to 5.2 billion tons by 2050 [2]. Each tone of clinker produced releases an average of 850 kg of carbon dioxide into the environment [3]. To reduce the negative impact of production cement on the environment, it is urgent to find alternative materials which can fully or partially replace cement. This can be achieved by using geopolymer concrete, partial replacement of cement with Supplementary Cementitious Materials (SCMs), and incorporation of nanomaterials for sustainability [4,5,6,7].

Numerous researchers have focused on the incorporation of SCMs such as fly ash and Ground Granulated Blast Furnace Slag (GGBFS) in concrete, which have an environmental affinity [8,9]. Depending on the chemical composition and the reactivity of SCMs, two groups can be mainly divided, namely hydration reactivity and pozzolanic reactivity. GGBFS is the representative material with hydraulic reactivity, and it contains high amorphous and calcium oxide content. As such, GGBFS can achieve a high replacement ratio of cement in the concrete mixture. Fly ash, meanwhile, is a typical material with the pozzolanic property, as it is rich in alumina and silica [10]. However, it is reported that the global annual production of fly ash is estimated to reach above 800 million tons [11]. Furthermore, although GGBFS production is more than 500 million tons per year, only around 65% of the total amount of GGBFS is reused, causing a slew of waste disposal issues, with the ashes being dumped in landfills or dumped in the ocean [12]. Therefore, it is critical to improve the utilisation efficiency of SCMs such as the use of the high volume of fly ash and GGBFS in concrete to reduce the waste of resources.

It has been reported that the addition of fly ash in concrete can alter the mechanical properties and time-dependent behaviours of concrete. Concrete mixes with fly ash at the content level of 50% can also provide advantages, including strong resistance to chloride and sulphate attack, less alkali–silica expansion, and low heat [13]. According to Moghaddam et al. [14], substituting OPC with fly ash increases porosity while decreasing average pore size. Furthermore, the volume of gel pore increases as the fly ash content rises. Saha [15] investigated concrete with fly ash and discovered that drying shrinkage reduces as the amount of fly ash in the mix increases. The addition of fly ash to concrete reduced its porosity, resulting in concrete with modified water sorptivity and chloride permeability. When substantial replacement volumes of fly ash were utilised, the resistance to chloride-ion penetration was also significantly increased [16]. Self-compacting concrete mixes with fly ash has also been shown to reduce autogenous and drying shrinkage [17]. However, due to the pozzolanic reaction of fly ash occurs slowly and the relatively low cement content in the concrete, the early strength of fly ash concrete is relatively lower compared to that of conventional concrete, resulting in a possible reduction of the load-carrying capacity of the member [18,19,20,21].

On the other hand, the report showed that adding GGBFS to concrete enhanced the long-term mechanical strength while also promoting the elastic modulus marginally. Furthermore, GGBFS can improve the concrete engineering properties by lowering shrinkage and creep as well as change the durability by enhancing abrasion resistance, reducing permeability, and greatly promoting the interactions between alkali–silica and sulfate [22]. In addition, Łukowski and Salih [23] observed that adding GGBFS to concrete can lower its porosity due to the larger surface area of GGBFS. It has also been stated that the concrete had a denser microstructure and was more solid regardless of the substitution rate [24]. Li et al. [25] studied the microstructure of cement paste containing GGBFS and discovered that GGBFS lowered pores volume significantly, increased specific surface area and fractal dimension, and lowered the diameter of pore from 10 to 100 nm to 10 nm. Self-compacting concrete containing GGBFS has also been reported to have a higher level of strength [26]. It has been observed that replacing up to 70% of the cement in self-consolidating concrete prepared with high GGBFS concrete had a 45% lower shrinkage than normal concrete [27]. Darquennes et al. [28] investigated the free shrinkage of concrete with 75% cement replaced by GGBFS. They discovered that the total shrinkage development rate was lower overtime for concrete with 50% cement replaced by GGBFS than conventional concrete. However, Zhao et al. [29] discovered that for high-performance concrete, the overall shrinkage containing fly ash and GGBFS was lowered by 15–25% compared to cement-only high-performance concrete. However, when fly ash and GGBFS were added to high-performance concrete, autogenous shrinkage increased by 66–106%. Therefore, it is crucial to evaluate the mechanical properties as well as autogenous shrinkage of concrete containing fly ash and GGBFS.

The mechanical properties of concrete at early age significantly affect the concrete cracking in the engineering field [30]. In existing standards, Eurocode 2 and FIB 2010 are commonly used prediction models to evaluate the early age mechanical properties of concrete [31,32]. However, the prediction model does not distinguish between conventional concrete and SCMs-based concrete. As a result, numerous researchers have developed various prediction models to investigate the time development of concrete mechanical properties. Singh et al. [33] created an artificial neural network (ANN) to predict the concrete compressive strength based on nonlinear regression analysis. Bhaskara et al. [34] adopted a reaction-kinetics-based strength development model to predict the strength development for OPC and fly ash concrete. Liu and Wang [35] proposed a model capturing the strength factor and maturity method to better reflect the strength development of fly ash concrete. However, owing to the complex procedures and more experimental parameters such as heat of hydration, these models are not suitable for implementation in existing codes. Therefore, it is important to examine the adaptability of existing models in predicting the time development of mechanical properties of concrete.

In this study, 12 concrete mixes including various fly ash and GGBFS content were cast to investigate the mechanical properties and autogenous shrinkage of concrete. Four different cement-replacement levels (0%, 20%, 30%, 40%) by fly ash and GGBFS were used to examine the influence of cement-replacement levels on the mechanical properties of concrete until 28 days. Three additional concrete mixes including ternary binders were also utilised to evaluate the early age concrete autogenous shrinkage. Prediction models in existing standards were utilised and compared to the time development of mechanical properties and autogenous shrinkage of concrete.

## 2. Materials and Methods

### 2.1. Materials and Mix Proportion

The raw materials in this study included Ordinary Portland cement (strength class 42.5) with a Blaine fineness of 375 m^2^/kg in accordance with Chinese Standard GB 175-2020 [36]. Two types of SCMs, namely fly ash and GGBFS, were used in experiments in accordance with Chinese Standard GB/T 1596-2017 [37] and GB/T 18046-2017 [38], respectively. Figure 1 presented the microstructure of fly ash and GGBFS using scanning electron microscopy (SEM) technology. Table 1 provided the chemical composition of cement, fly ash, and GGBFS using X-ray fluorescence (XRF). Two kinds of coarse aggregate were used including the crushed stone with a maximum nominal size of 20 mm and 10 mm, respectively. The ratio of 20 mm coarse aggregate to 10 mm coarse aggregate is 1.5:1 by weight. Fine aggregate was manufactured sand with a maximum aggregate size of 2.6 mm. Polycarboxylate high-range water reducer was utilised to improve the workability of the mixtures in this study.

The mix proportion of concrete in this paper is shown in Table 2. A total of 12 groups of concrete mixtures were designed. For the first nine groups, a fixed w/b ratio of 0.38 was used, and they were used to examine the influences of fly ash and GGBFS on the development of the mechanical properties of the concrete. For the last three groups, different w/b ratios and SCMs replacement levels were used to investigate the effect of ternary binders on the autogenous shrinkage of the concrete. Four different cement replacement levels, 0%, 20%, 30%, and 40%, by fly ash and GGBFS and ternary binders were adapted. According to different proportions of fly ash and GGBFS, the concrete mixtures were designated as ‘C100’, ‘C80F20’, ‘C80S20’, ‘C60F20S20’, etc. Mixture ‘C100’ was the reference mixture mixes without SCMs, while ‘C80F20’ and ‘C80S20’ were the concrete mixes with 20% fly ash and 20% GGBFS as a cement replacement by weight, respectively. ‘C60F20S20’ represented concrete mixes with ternary binders (20% fly ash and 20% GGBFS) were used to replace cement.

### 2.2. Mechanical Properties Investigation

The mechanical properties including compressive strength, splitting tensile strength, flexural strength, and elastic modulus of concrete for all mixes were conducted according to Chinese Standard GB/T 50081-2019 [39]. The compression and splitting tensile strength were tested using a hydraulic universal testing machine as shown in Figure 2. Cubic specimens were cast for determining the compressive strength and splitting tensile strength. The typical size of cubic specimen is 150 mm × 150 mm × 150 mm. Prismatic specimens with a dimension of 150 mm × 150 mm × 600 mm were cast for flexural strength, and 150 mm × 150 mm × 300 mm concrete prisms were cast for determining the elastic modulus, respectively. The specimens were cured in standard conditions at a temperature of 20 °C ± 1 °C and relative humidity of 95 ± 5%. The mechanical properties tests were carried out at the age of 2, 3, 7, and 28 days. In addition, the workability of concrete was evaluated by slump test and the slump of all concrete mixtures was controlled as 180 ± 20 mm.

### 2.3. Autogenous Shrinkage Measurement

The size of autogenous shrinkage specimens is 100 mm × 100 mm × 400 mm. The specimens were demoulded at 24 h after casting and stored in an environmentally controlled room at a temperature of 20 °C ± 2 °C and RH of 60 ± 5%. As shown in Figure 3, the specimens were immediately wrapped with the plastic film after demoulding to avoid moisture loss. Thermocouples were embedded in the specimens to monitor the change of the internal temperature. A shrinkage dial gauge was utilised to measure the autogenous shrinkage of concrete. The readings were recorded at 1, 3, 7, 14, and 28 days after the benchmark reading.

## 3. Results and Discussion

### 3.1. Mechanical Properties of Concrete

Figure 4 presents the development of compressive strength for all concrete mixes. An average of 3 specimens was used for each displayed point. As shown in Figure 4a, it was observed that for concrete mixes with fly ash, the compressive strength was about 30% lower than that of the control mix at an early age. It also observed that the higher fly ash content in concrete, the lower the compressive strength of concrete at an early age. Similar results can be found in [40,41]. The reason can be attributed to the dilution effect on cement hydration affected by the addition of fly ash [40]. Moreover, the pozzolanic reaction between fly ash particles and calcium hydroxide also retarded the hydration degree [42]. The rate of compressive strength development of fly ash concrete was higher than that of OPC mix at a later age. Excluding the C80F20 mix, the compressive strength of fly ash concrete was only marginally lower than that of the reference mix at 28 days. This can be attributed to the pozzolanic reaction leading to an ever-increasing reaction degree. Fly ash particles can also serve as nucleation sites to accelerate the hydration of cement, leading to an insignificant difference from that of the control mix [43,44]. As shown in Figure 4b, it was observed that the compressive strength of GGBFS concrete was lower than that of the OPC mix at 2 and 3 days (early age). The higher the GGBFS content in concrete, the lower the compressive strength. At 3 days, the concrete mixes with the 20%, 30%, and 40% GGBFS were observed to be 74%, 71%, and 60%, respectively, of that of plain concrete. Similar results were reported by Shariq et al. [45]. However, at the age of 28 days, the compressive strength of GGBFS concrete was 11%, 17%, and 8% higher, respectively, than that of control concrete. This was attributed to the late pozzolanic reactivity of GGBFS contributes to the rate of strength gain at later ages [46]. For the ternary binder group, the concrete mixes with fly ash and GGBFS were lower than reference concrete at all different ages. Such a behaviour was also noted by Wang and Chen [47]. In addition, the measured compressive strength for autogenous shrinkage specimens including C65F35, C60F25S15, and C65F15S20 were 30.1, 40.3, and 45.5 MPa at 28 days.

Figure 5 showed the development of the splitting tensile strength of concrete. For concrete mixes with fly ash, the splitting tensile strength at the age of 28 days after casting for C80F20, C70F30, and C60F40 were 5.35, 4.93, and 4.02 MPa, respectively. It was seen that the higher fly ash content, the lower the average value of splitting tensile strength at 28 days. Such behaviour was also noted by Zhang et al. [48]. For the GGBFS group, the average value of splitting tensile strength of GGBFS concrete at 28 days was decreased by 15%, 18%, and 21% compared to that of OPC mix when the GGBFS content increased from 0% to 20%, 30%, 40%, respectively. Similar results were obtained by Shen et al. [49]. For concrete mixes with ternary binders, it was observed that the higher ternary binders content results in a lower splitting tensile strength. The reason was attributed to the water required for hydration being insufficient, leading to the reduction of the hydration products [50].

The development of flexural strength of concrete was provided in Figure 6. It can be seen from Figure 6 that the introduction of fly ash decreased the flexural strength at 28 days. As the replacement of fly ash increased, the difference between fly ash concrete and reference concrete in flexural strength was larger at 28 days. Results on flexural strength of fly ash concrete were in accordance with the results in [51]. A similar trend was observed for GGBFS and ternary binder group. It was reported that GGBFS improved the early age flexural strength, but GGBFS also reduced the calcium ion concentration between cement and aggregate, leading to a decrease in flexural resistance at a later age [52].

The 7- and 28-day elastic modulus of concrete was provided in Table 3. The elastic modulus was calculated from the compressive test. It was observed that fly ash and GGBFS can improve the 7-day elastic modulus by about 3% to that of the control mix. While the ternary binders lead to a 19% decrease in elastic modulus at 7 days. For 28 days results, it can be seen that the addition of fly ash, GGBFS, and ternary binders results in a marginally reduction of 5%, 10%, and 10% lower than that of the reference mix. Similar test results can be found in [53,54].

### 3.2. Autogenous Shrinkage of Concrete

Figure 7 showed the autogenous shrinkage of concrete mixes with ternary binders. As mentioned in the previous section, three different concrete mixes with different w/b ratios and cement replacement are cast to investigate the autogenous shrinkage. It should be noted that the autogenous shrinkage is related to the compressive strength of concrete. As a result, it is difficult to compare the autogenous shrinkage of concrete with different SCMs replacement levels and compressive strength. However, it was reported that the autogenous shrinkage of fly ash and GGBFS concrete was higher compared to that of OPC concrete due to pore structure refinement [55,56]. This point was also supported by this study. It was clear that despite the compressive strength of C60F25S15 being higher than C65F35, the autogenous shrinkage of C60F25S15 was lower than that of C65F35, indicating the incorporation of fly ash and GGBFS increased the autogenous shrinkage of concrete which compensated the effect of compressive strength.

## 4. Assessment of Models Predicting Mechanical Properties and Autogenous Shrinkage of Concrete

### 4.1. Mechanical Properties of Concrete

The development of the compressive strength of concrete in Eurocode 2 and FIB 2010 models [31,32] is expressed as a function of time and compressive strength at 28 days, as shown in Equation (1):(1)$$f_{c}\left(t\right)=f_{c,28}\exp\left[s_{1}\left(1-\sqrt{\frac{28}{t}}\right)\right]$$
where $s_{1}$ depends on the cement type and is equal to 0.2, 0.25, and 0.38 for cement strength classes of 32.5N, 42.5N, and 52.5N, respectively.

Figure 8 showed a comparison between predicted and measured development of compressive strength using $s_{1}=0.25$ for all concrete mixes with SCMs. It was obvious that the prediction model cannot well predict the development of compressive strength for all concrete mixes. The estimated development of compressive strength was lower than that of experimentally measured values. As a result, the value of $s_{1}$ should be reconsidered when the fly ash and GGBFS were presented in concrete.

The development of the splitting tensile strength of concrete in Eurocode 2 and FIB 2010 models [31,32] is expressed as a function of time and splitting tensile strength at 28 days, as shown in Equation (2):(2)$$f_{sp}\left(t\right)=f_{sp,28}\exp\left[s_{2}\left(1-\sqrt{\frac{28}{t}}\right)\right]$$
where $s_{2}$ depends on the cement type and is equal to 0.2, 0.25, and 0.38 for cement strength classes of 32.5N, 42.5N, and 52.5N, respectively.

Figure 9 displayed the prediction of the splitting tensile strength development based on $s_{2}=0.25$ for all concrete mixes with SCMs. It can be seen that the prediction of time-dependent splitting tensile strength was different from that of the prediction of compressive strength. The predicted values were higher than that of the experimental values. As such, the value of $s_{2}$ in the prediction model also needs to be reconsidered.

The development of the elastic modulus of concrete in Eurocode 2 and FIB 2010 models [31,32] is expressed as a function of time and elastic modulus at 28 days, as shown in Equation (3):(3)$$E_{c}\left(t\right)=E_{c,28}\exp\left[s_{3}\left(1-\sqrt{\frac{28}{t}}\right)\right]$$
where $s_{3}$ depends on the cement type and is equal to 0.2, 0.25, and 0.38 for cement strength classes of 32.5N, 42.5N, and 52.5N, respectively.

Figure 10 presented the development of experimental values of elastic modulus and the values from prediction models based on $s_{3}=0.25$ for concrete mixes with SCMs. It can be seen that the prediction values were close to experimental values for fly ash and GGBFS mixes because the early age elastic modulus results are unknown. However, the prediction results were overestimated for the ternary binder group. Therefore, the value of $s_{3}$ is recalculated to fit the experimental results.

The incorporation of SCMs can influence prediction results as per [31,32,57]. As such, Table 4 displayed the values of fitting parameters based on the experimental results using the Levenberg–Marquardt least square regression analysis method. It can be seen that the fitting parameters values of $s_{1}$, $s_{2}$, and $s_{3}$ for different groups were different. The average values of fitting parameters $s_{1}$, $s_{2}$, and $s_{3}$ were 0.28, 0.48, 0.23 and 0.34, 0.39, 0.15 and 0.35, 0.40, 0.45 for fly ash group, GGBFS group, and ternary binder group, respectively. It also obvious that these values were different from the suggested values in standards [31,32]. Therefore, it is recommended to consider the parameter capturing the effect of fly ash and GGBFS on concrete mechanical properties development.

### 4.2. Autogenous Shrinkage of Concrete

Regarding autogenous shrinkage of concrete mixes with SCMs, three empirical models from standards are assessed in this study [31,32,58]. The reason to select these prediction models is due to the simplicity and wide-range application in the industry. The prediction in these models is a strength-based approach rather than focusing on the mix design, which is widely used in design engineering field.

The expression of autogenous shrinkage of concrete in Eurocode 2 [31] is shown as follows:(4)$$\varepsilon_{au}\left(t\right)=\varepsilon_{au}^{*}\times \left(1-e^{-0.2\sqrt{t}}\right)$$
where $\varepsilon_{au}^{*}$ is the final autogenous shrinkage which is calculated as Equation (5):(5)$$\varepsilon_{au}^{*}=2.5\times \left(f_{c,28}-10\right)\times 10^{-6}$$
where $f_{c,28}$ is the characteristic compressive strength of concrete.

The autogenous shrinkage in FIB 2010 [32] is estimated by means of basic notional shrinkage coefficient and the time function, as shown in Equations (6) and (7):(6)$$\varepsilon_{au}\left(t\right)=\varepsilon_{au}^{*}\times \left(1-e^{-0.2\sqrt{t}}\right)$$
(7)$$\varepsilon_{au}^{*}=\alpha_{bs}\times {\left(\frac{0.1f_{c,28}}{6+0.1f_{c,28}}\right)}^{2.5}\times 10^{-6}$$
where $\alpha_{bs}$ depends on the cement type and is equal to 800, 700 and 600 for cement strength class of 32.5N, 42.5N and 52.5N, respectively.

In AS3600-2018 [58], the autogenous shrinkage strain is calculated as follows:(8)$$\varepsilon_{au}\left(t\right)=\varepsilon_{au}^{*}\times \left(1.0-e^{-0.07t}\right)$$
(9)$$\varepsilon_{au}^{*}=\{\begin{array}{c} \left(0.07f_{c,28}-0.5\right)\times 50\times 10^{-6}\\ \left(0.08f_{c,28}-1.0\right)\times 50\times 10^{-6} \end{array}\begin{array}{c} for\;f_{c,28}\leq 50\;MPa\\ for\;f_{c,28}>50\;MPa \end{array}$$

Figure 11 showed a comparison between experimental values of autogenous shrinkage and the predicted values for concrete mixes with SCMs. It can be seen that AS3600-2018 model provides the most accurate prediction for C60F25S15 ($f_{c,28}=40.3$ MPa), while the Eurocode 2 and FIB 2010 models exhibited poor estimations. However, for C65F35 ($f_{c,28}=30.1$ MPa) and C65F15S20 ($f_{c,28}=45.5$ MPa) concrete, although the AS3600-2018 model showed a better prediction than others, the predicted values were still lower than experimental autogenous shrinkage values. Therefore, it is recommended that the prediction models in these standards should take account into the effect of fly ash and GGBFS on autogenous shrinkage of concrete.

## 5. Modelling Development of Mechanical Properties of Concrete with Fly Ash and GGBFS

### 5.1. Proposed Model

As mentioned in the previous section, the effect of fly ash and GGBFS on concrete mechanical properties development should be considered in existing models. As shown in Table 4, a linear relationship can be observed for regression coefficients $s_{1}$ and $s_{2}$. However, due to insufficient reliable data of $s_{3}$, the modeling of $s_{3}$ was not presented in this study. Therefore, the model capturing the effect of fly ash and GGBFS was proposed for $s_{1}$ and $s_{2}$ as follows:(10)$$s_{1}=s_{1,0}\times m_{1}\times n_{1}$$
(11)$$s_{2}=s_{2,0}\times m_{2}\times n_{2}$$
where $s_{1,0}$ and $s_{2,0}$ are the recommended parameters for concrete without fly ash and GGBFS, and the recommended values in this study are 0.146 and 0.389 for $s_{1,0}$ and $s_{2,0}$, respectively; $m_{1}$ and $n_{1}$ are the factor depending on the percentage of fly ash and GGBFS for $s_{1}$, as shown in Equations (12) and (13); $m_{1}$ and $n_{2}$ are the factor depending on the percentage of fly ash and GGBFS for $s_{2}$, as shown in Equations (14) and (15).
(12)$$m_{1}=0.08\times FA-0.46$$
(13)$$n_{1}=0.009\times GGBFS+2.08$$
and
(14)$$m_{2}=-0.026\times FA+2.03$$
(15)$$n_{2}=-0.006\times GGBFS+1.19$$
where FA and GGBFS are the percentage of fly ash and *GGBFS* used in concrete. It should also be noted that the Equations (10) and (11) can be used not only for concrete with fly ash or *GGBFS*, but also extend to concrete with ternary binders.

### 5.2. Proposed Model Comparison with Experimental Results

Figure 12 and Figure 13 show the comparison of development of compressive strength and splitting tensile strength and the predicted results using the proposed model. It can be observed that the predicted values agreed well with the experimentally measured results. This indicates that the proposed model for calculating parameters $s_{1}$ and $s_{2}$, which captured the effect of fly ash and GGBFS, could be used in the existing models.

The evaluation and error values have been widely interpreted in many studies [59,60,61]. The mean value and coefficient of variation (CoV) are typically used to indicate the error and robustness. As shown in Figure 14, the mean value and CoV of the ratio by the proposed model and experimental results are 1.00, 0.10, and 0.95, 0.17 for the $s_{1}$ and $s_{2}$ prediction models, respectively. As a result, it can be seen that the statical analysis confirms the proposed model has been successfully utilised in the existing models to calculate the time-dependent compressive strength and splitting tensile strength of concrete.

## 6. Conclusions

The results presented in this work provide insight into the mechanical properties and autogenous shrinkage development of concrete mixes with various fly ash and GGBFS content. The observed trends and patterns of behaviour could be utilised to reflect the early-age behaviour of concrete. The main observations are as follows:Fly ash and GGBFS reduced the early age mechanical properties of concrete, but the compressive strength and splitting tensile strength of fly ash concrete were similar to OPC concrete at 28 days, while GGBFS increased the compressive strength of concrete at 28 days. Both fly ash and GGBFS decreased the flexural strength and elastic modulus of concrete. When ternary binders were incorporated, the concrete also exhibited low mechanical properties.The model in existing standards overestimated the early age of compressive strength development but underestimated the splitting tensile strength of concrete, leading to an unreliable calculation for design purposes. As such, it is recommended to consider the effect of fly ash and GGBFS on concrete’s mechanical properties development.A model capturing the effect of fly ash and GGBFS on the development of compressive and splitting tensile strength was proposed. The statistical analysis of the comparison between predicted and measured compressive strength and splitting tensile strength showed a good agreement. The mean value and CoV are 1.00, 0.10, and 0.95, 0.17 for the $s_{1}$ and $s_{2}$ prediction models, respectively.The autogenous shrinkage of concrete mixes with fly ash and GGBFS was different from that of OPC concrete. The model in existing standards failed in predicting the early-age autogenous shrinkage development of fly ash and GGBFS concrete. As a result, a broader database of concrete containing fly ash and GGBFS values is recommended to improve and establish optimal parameters in existing models.

## Figures and Tables

**Figure 1 materials-15-03016-f001:**
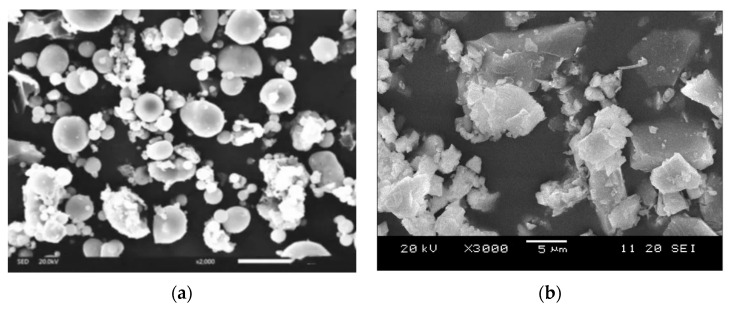
SEM graph of SCMs: (**a**) fly ash; (**b**) GGBFS.

**Figure 2 materials-15-03016-f002:**
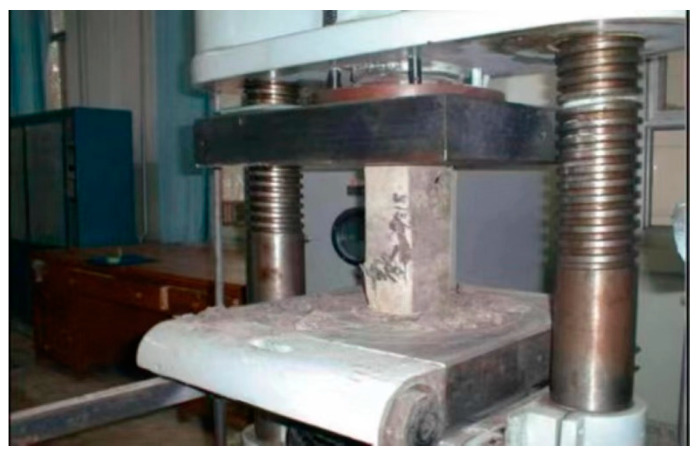
Concrete specimens and test apparatus.

**Figure 3 materials-15-03016-f003:**
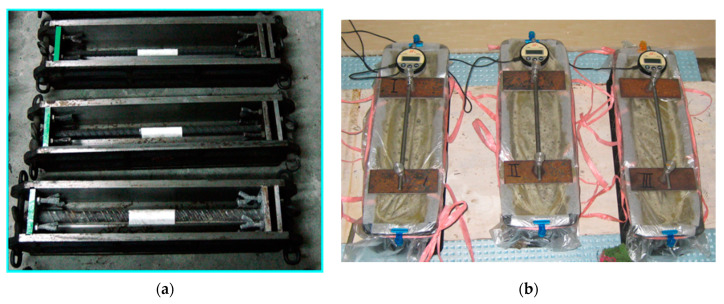
Autogenous shrinkage of concrete: (**a**) shrinkage mould; (**b**) specimens.

**Figure 4 materials-15-03016-f004:**
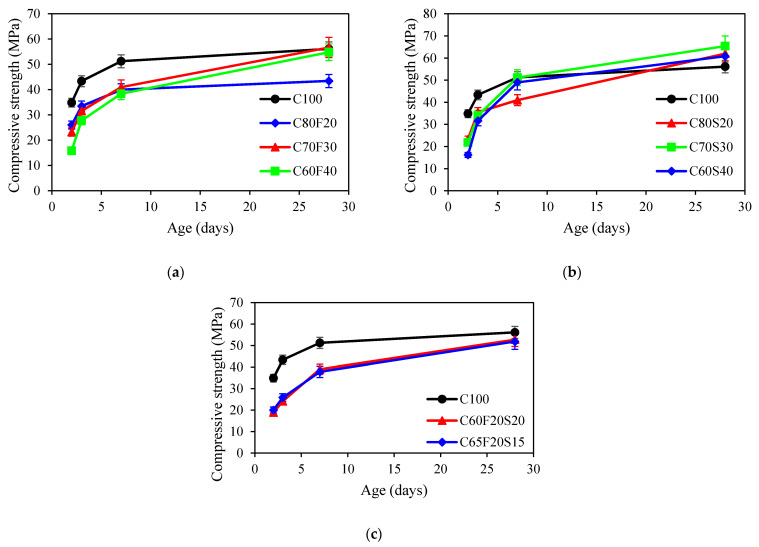
Compressive strength of concrete: (**a**) fly ash group; (**b**) GGBFS group; (**c**) ternary binder group.

**Figure 5 materials-15-03016-f005:**
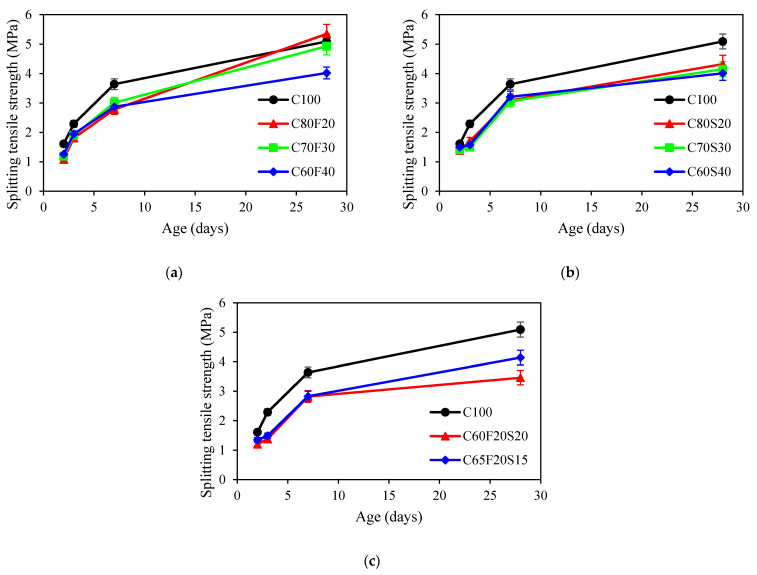
Splitting tensile strength of concrete: (**a**) fly ash group; (**b**) GGBFS group; (**c**) ternary binder group.

**Figure 6 materials-15-03016-f006:**
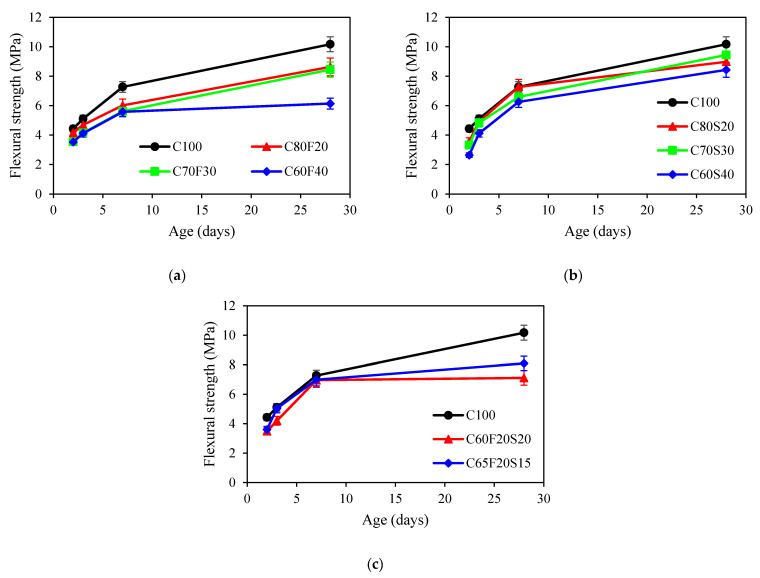
Flexural strength of concrete: (**a**) fly ash group; (**b**) GGBFS group; (**c**) ternary binder group.

**Figure 7 materials-15-03016-f007:**
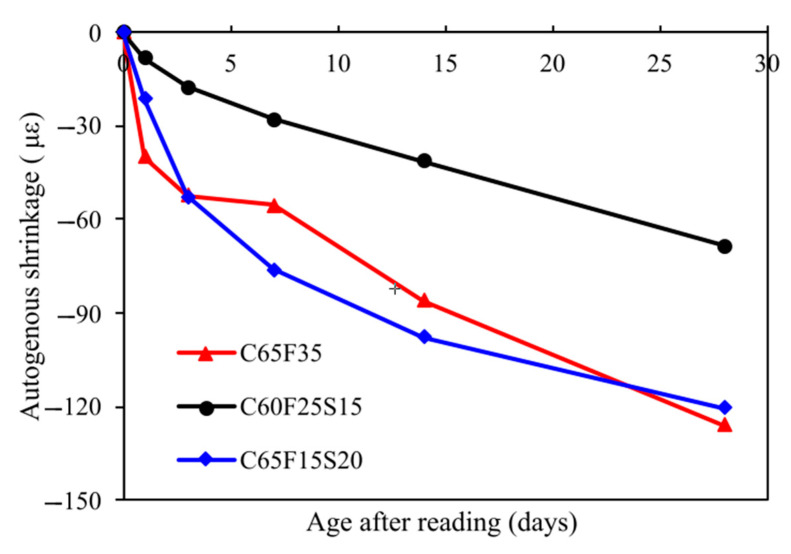
Autogenous shrinkage of concrete.

**Figure 8 materials-15-03016-f008:**
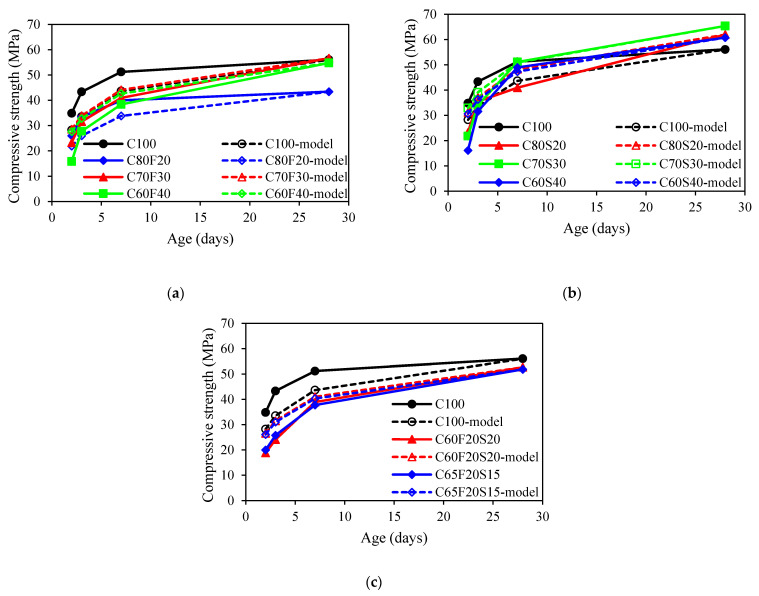
Comparison of the compressive strength development between experimental results and model predictions: (**a**) fly ash group; (**b**) GGBFS group; (**c**) ternary binder group.

**Figure 9 materials-15-03016-f009:**
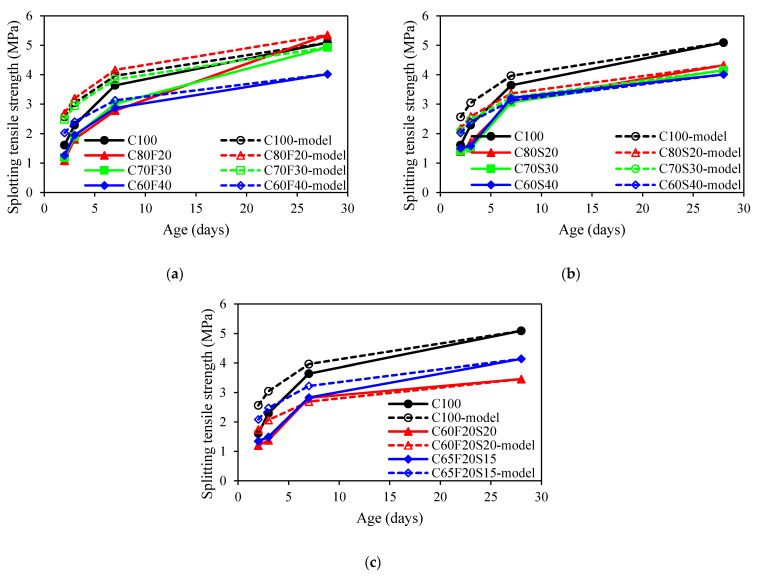
Comparison of the splitting tensile strength development between experimental results and model predictions: (**a**) fly ash group; (**b**) GGBFS group; (**c**) ternary binder group.

**Figure 10 materials-15-03016-f010:**
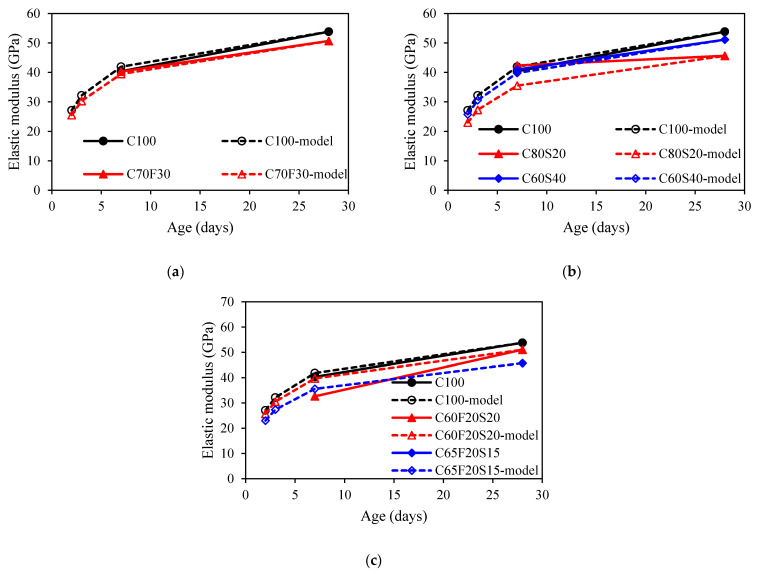
Comparison of the elastic modulus development between experimental results and model predictions: (**a**) fly ash group; (**b**) GGBFS group; (**c**) ternary binder group.

**Figure 11 materials-15-03016-f011:**
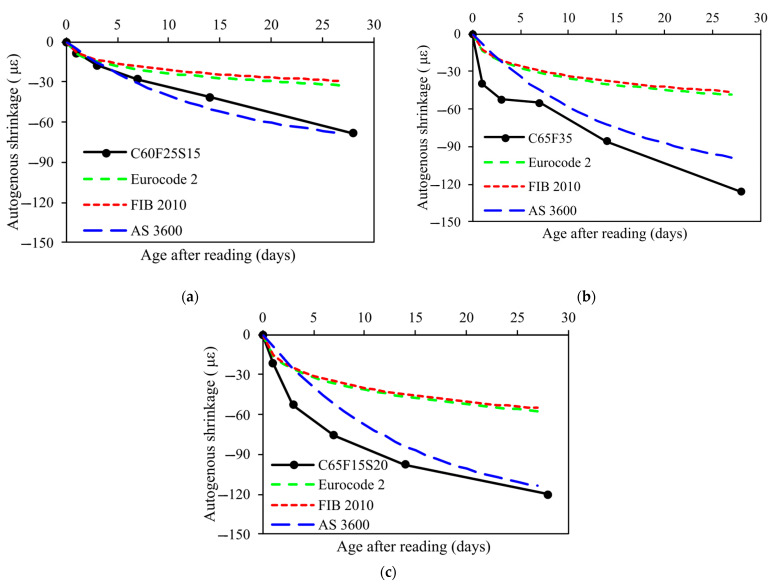
Comparison of the autogenous shrinkage development between experimental results and model predictions: (**a**) C60F25S15; (**b**) C65F35; (**c**) C65F15S20.

**Figure 12 materials-15-03016-f012:**
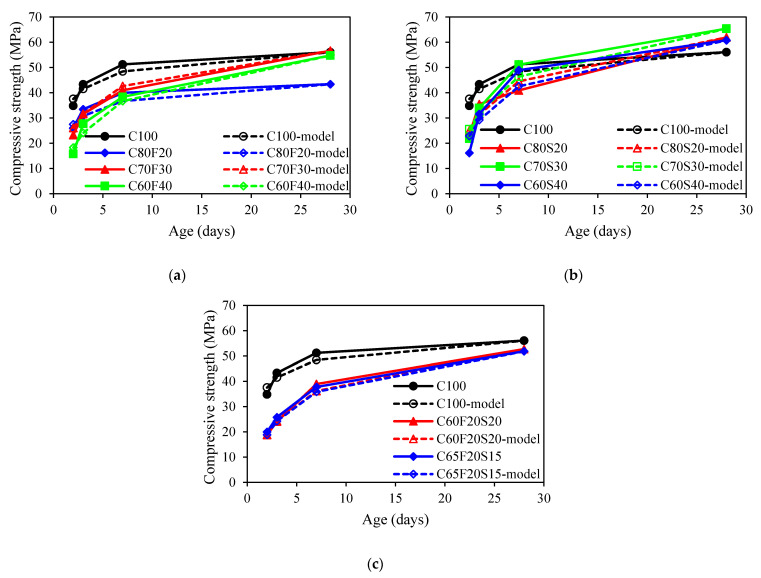
Comparison between experimental data and predicted compressive strength using proposed models for *s*_1_: (**a**) fly ash group; (**b**) GGBFS group; (**c**) ternary binder group.

**Figure 13 materials-15-03016-f013:**
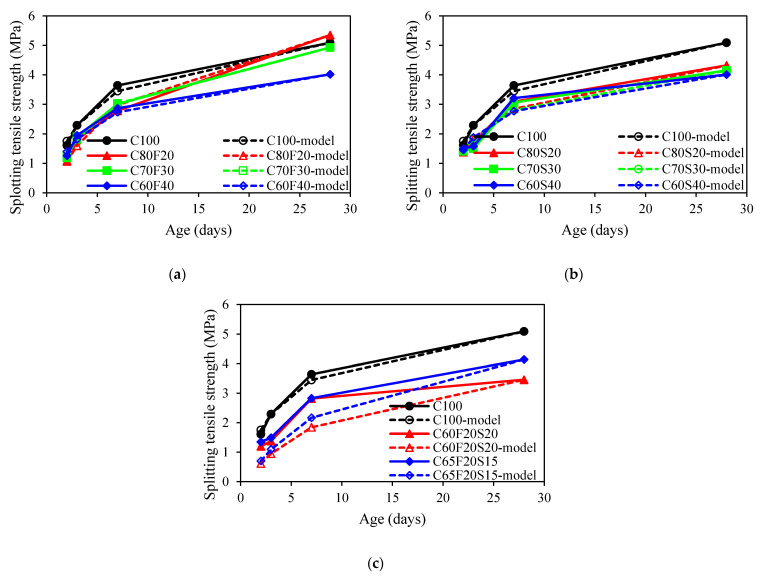
Comparison between experimental data and predicted splitting tensile strength using proposed models for *s*_2_: (**a**) fly ash group; (**b**) GGBFS group; (**c**) ternary binder group.

**Figure 14 materials-15-03016-f014:**
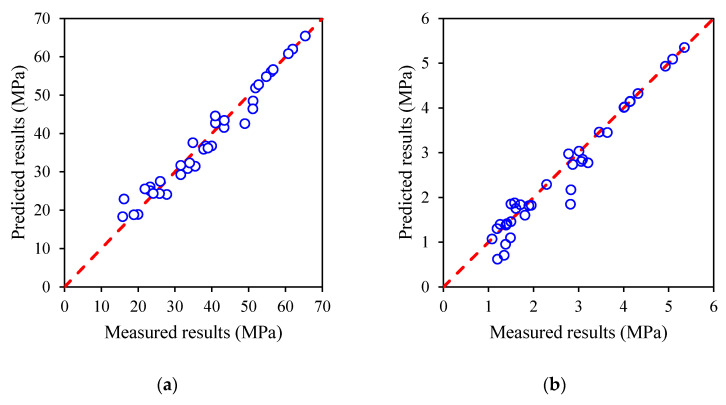
Comparison between experimental data and predicted results using proposed models for: (**a**) compressive strength with parameter *s*_1_; (**b**) splitting tensile strength with parameter *s*_2_.

**Table 1 materials-15-03016-t001:** Chemical compositions of cement, fly ash, and GGBFS (wt.%).

Chemical Composition	Cement	Fly Ash	GGBFS
CaO	63.22	1.93	34.69
MgO	1.56	0.95	7.21
Fe_2_O_3_	3.25	4.88	1.86
Al_2_O_3_	3.67	33.18	15.55
SiO_2_	22.89	55.91	36.93
SO_3_	2.09	0.51	0.66
Loss on ignition (LoI)	2.14	0.99	2.19

**Table 2 materials-15-03016-t002:** Mix proportions of concrete by weight.

Sample ID	Mix Proportions (kg/m^3^)						
	Cement	FA	GGBFS	Coarse Agg ^1^	Coarse Agg ^2^	Fine Agg	Water
C100	400	0	0	439	658	731.2	152
C80F20	320	80	0	439	658	731.2	152
C70F30	280	120	0	439	658	731.2	152
C60F40	240	160	0	439	658	731.2	152
C80S20	320	0	80	439	658	731.2	152
C70S30	280	0	120	439	658	731.2	152
C60S40	240	0	160	439	658	731.2	152
C65F20S15	260	80	60	439	658	731.2	152
C60F20S20	240	80	80	439	658	731.2	152
C65F35	247	133	0	146	292	780	180
C60F25S15	228	95	57	146	292	747	152
C65F15S20	292	68	90	146	292	778	155

^1^ Coarse aggregate with max aggregate size of 10 mm. ^2^ Coarse aggregate with max aggregate size of 20 mm.

**Table 3 materials-15-03016-t003:** Elastic modulus of concrete (GPa).

Sample ID	7d	28d
C100	40.37	53.82
C80F20	43.01	N/A
C70F30	40.29	50.67
C80S20	42.33	45.62
C60S40	40.97	51.12
C65F20S15	N/A	45.77
C60F20S20	32.68	51.09

**Table 4 materials-15-03016-t004:** Regression coefficients of all mixtures.

Sample ID	$s_{1}$	$s_{2}$	$s_{3}$
C100	0.146	0.389	0.288
C80F20	0.152	0.582	N/A
C70F30	0.309	0.49	0.229
C60F40	0.382	0.377	N/A
C80S20	0.332	0.41	0.075
C70S30	0.337	0.41	N/A
C60S40	0.357	0.363	0.221
C65F20S15	0.339	0.434	N/A
C60F20S20	0.363	0.368	0.447

## Data Availability

Data sharing is not applicable to this article.

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
