# Peer review of "Effect of Various Fly Ash and Ground Granulated Blast Furnace Slag Content on Concrete Properties: Experiments and Modelling"

_materials, 2022, doi:10.3390/ma15093016_

Round 1

Reviewer 1 Report

‎           The paper of (Effect of Various Fly Ash and GGBFS Content ‎on ‎Concrete Properties: Experiments and Modelling) was well ‎written. However, it needs to improve the quality by addressing some ‎comments. So ‎need to follow and correct comments and mistakes

‎1.‎         Abstract should be shown the introduction, problem, aim, and scope ‎of the study, method, and results. In addition to a conclusion, and some future ‎recommendations.‎

‎2.‎         Keywords should represent the most important and more repeated ‎words, however, the word ((Mechanical property)) did not ‎repeat in the overall paper. Therefore, it should be converted to ‎‎((Mechanical properties)).‎

‎3.‎         In The first paragraph in the introduction part, please insert  ‎suitable and newest references for the sentence (Numerous ‎researchers focused on the incorporation of Supplementary ‎Cementitious Materials (SCMs) such as fly ash and Ground ‎Granulated Blast Furnace Slag (GGBFS) in concrete, which has ‎an environmental affinity).‎

‎4.‎         Please insert the newest citations in the introduction part instead of the old ones.

‎5.‎         Please note that the mechanical properties include the ‎compressive, flexural, and splitting tensile strengths, elastic ‎modulus, Toughness, Hardness, Hardenability, Brittleness, ‎Malleability, Ductility, Creep, Slip, etc..‎

‎6.‎         In Table 1, the summation of the chemical composition of Cement, Fly ‎ash, and GGBS should be equal to 100% for each material.

‎7.‎         Please rewrite the sentence of (While for GGBFS concrete as ‎shown in Figure 4 (b), it was observed that the compressive strength ‎of GGBFS concrete was also lower than that of OPC mix at an ‎early age. Similarly, it was also seen that the compressive strength ‎was increasing with the GGBFS content from 20% to 40%) located ‎between lines 183 to 186 to be more clear.‎

‎8.‎         More results obtained in this study should be stated in the ‎conclusion part.‎

  1.  Plagiarism rate is high, especially at section 2.2 and other sections.

Reviewer 2 Report

In this manuscript, four different cement replacement levels 15 including 0%, 20%, 30%, and 40% by fly ash and GGBFS as well as ternary binders were considered as Supplementary Cementitious Materials (SCMs) and the concrete properties including these materials were investigated experimentally and numerically.

the experimental mechanical properties and their corresponding numerical simulation of C80concrete were investigated. The manuscript is well written and is of great importance in the concrete field of research. Some questions and suggestions are provided below to enrich the scientific level of the presented manuscript before its publication:

  • It is suggested to add some figures of supplementary cementitious materials (fly ash and GGBFS ) to the materials section.
  • It would enrich the scientific level of manuscript if the authors compute the evaluation and error values of presented results in figures 3, 4 and 5 based on existing empirical models. The same procedure can be conducted on figures 7 to 10. The following papers can be considered as the reference:
    • Faridmehr, I., Shariq, M., Plevris, V., & Aalimahmoody, N. (2022). Novel hybrid informational model for predicting the creep and shrinkage deflection of reinforced concrete beams containing GGBFS. Neural Computing and Applications, 1-17.
    • Jahangir, H., & Eidgahee, D. R. (2021). A new and robust hybrid artificial bee colony algorithm–ANN model for FRP-concrete bond strength evaluation. Composite Structures257, 113160.
    • Petrounias, P., Rogkala, A., Giannakopoulou, P. P., Christogerou, A., Lampropoulou, P., Liogris, S., ... & Koukouzas, N. (2022). Utilization of Industrial Ferronickel Slags as Recycled Concrete Aggregates. Applied Sciences12(4), 2231.
    •  
  • It is suggested to summarize the conclusions as bullets.

Reviewer 3 Report

The subject of this manuscript is well framed in the scientific domain of the Materials Journal. Although numerous works have been published in this domain, this still deserves further research. In general, the manuscript is well written. In fact, the manuscript is mainly divided into two parts. The first part covers the experimental study to investigate the time-dependent development of mechanical properties and autogenous shrinkage of cementitious materials containing various contents of fly ash and ground granulated blast furnace slag. The second part which seems to be the novelty of this study tries to predict the abovementioned properties using regression analysis of the existing theoretical/empirical models. However, the main contribution of this part is still not scientifically grown and needs further research. The following comments should be addressed to add value to the manuscript:

In the Title: In the current format of the manuscript, the word “modelling” in the title seems unjustified. The authors used previous models to predict the properties without any efforts to change the current models. This, however, can be justified by the following suggestions of the reviewer.

In Abstract; Lines 19-23: Authors stated that “In addition, prediction models in existing standards were used and compared to experimental results. The comparison results showed that the prediction models overestimated the compressive strength but underestimated the splitting tensile strength development and autogenous shrinkage. As a result, the prediction models need to be improved for current standards and empirical models.” So, how can one improve these models? Do the authors have any suggestions? For example, adding new terms to current models that take into account the effects of supplementary cementitious materials (SCMs) on the investigated properties.

In Introduction; Lines 28-29: Concrete is known as the most globally used construction material because of its excellent performance of mechanical properties and workability.This statement is not always true. Actually, concrete exhibits low tensile strength and fracture toughness, resulting in cracking. Therefore, rebar and more recently fibers are added to the cement matrix to increase the tensile properties of the concrete. This should be mentioned in the manuscript. See this reference to discuss this: "Design and Predicting performance of carbon nanotube reinforced cementitious materials: mechanical properties and dispersion characteristics."

In Introduction; Lines 35-37: To reduce the negative impact of production cement on the environment, it is urgent to find alternative materials which can fully or partially replace cement.” Please mention various routes for doing so; e.g., geopolymer concrete, partially replacement of cement with SCMs, and incorporation of nanomaterials for sustainability. Some suggested references are: "Modeling the mechanical properties of cementitious materials containing CNTs", "Probabilistic model for flexural strength of carbon nanotube reinforced cement-based materials", "Elastic modulus formulation of cementitious materials incorporating carbon nanotubes: Probabilistic approach", and "Probabilistic model for flexural strength of Cementitious Materials Containing CNTs."

In Introduction; Lines 93-94: “The early age mechanical properties of concrete significantly affect the concrete cracking in the engineering field.” Nanofiber/tubes could be used to prevent/delay the initiation of cracks. See this reference to discuss: "Mechanical properties of carbon-nanotube-reinforced cementitious materials: database and statistical analysis."

In Materials and Methods; Table 2: What is GP cement?

In Materials and Methods; Lines 153-154: “The mechanical properties tests were carried out at the age of 2, 3, 7, and 28 days.” Should it be the age of 1-day or 2-day? Please confirm this.

In Results and Discussions; Line 171: “Three specimens were used for each displayed point.” It is better to write “the average of 3 specimens was used.” Also, I suggest to add the error bars in the figures.

In Results and Discussions; Lines 173-174: Authors stated that “It also observed that the higher fly ash content in concrete, the lower the compressive strength of concrete.” This is only true at early ages. At 28-days, this statement is not true.

In Results and Discussions; Lines 181-183: “Fly ash particles can also serve as nucleation sites to accelerate the hydration of cement, leading to an insignificant difference from that of the control mix [36].” Similar results have also been found when incorporating nanomaterials due to their high surface energy. Please discuss this phenomenon. See this reference to discuss: "Carbon nanotube reinforced cementitious composites: A comprehensive review."

In Results and Discussions; Lines 204-205: “It was seen that the higher fly ash content, the lower the splitting tensile strength.” and Lines 206-209: “While for the GGBFS group, unlikely to compressive strength, the splitting tensile strength of GGBFS concrete at 28 days was decreased by 15%, 18%, and 21% compared to that of OPC mix when the GGBFS content increased from 0% to 20%, 30%, 40%, respectively.” This is only true for the average values at the age of 28-days. I think if you show the error bars, this is not true from the statistical point of view.

In Results and Discussions; Lines 219-220: “It can be seen from Table 5 that the introduction of fly ash decreased the flexural strength.” In this sentence, “Table 5” should be replaced with “Figure 5.”

In Results and Discussions; Lines 220-222: “As the replacement of fly ash increased, the difference between fly ash concrete and reference concrete in flexural strength was larger.” Again, this may not be significant statistically. It's only obvious at 28-day between C60F40 and the other two mixes.

In Results and Discussions; Line 231: “The 7- and 28-day elastic modulus of concrete was provided in Table 3.” The elastic modulus was calculated from the flexural test or the compressive test? Please mention this.

In Assessment of models predicting mechanical properties and autogenous shrinkage of concrete (Section 4): The authors stated in Lines 265-266, Lines 280-281, Lines 296-297 that the values of S1, S2, and S3 should be reconsidered. Also, in Lines 334-336 it was stated that “Therefore, it is recommended that the prediction models in these standards should take account into the effect of fly ash and GGBFS on autogenous shrinkage of concrete.” Is it possible that the authors introduce a new parameter to take into account these effects? Also, the prediction models seem to not working even for the control specimens without SCMs. How do the authors explain this?

In Table 4. Regression coefficients of all mixtures: I think it is better that the authors illustrate the experimental vs. prediction model using these corrected coefficients in the Figure. Also, is it possible to add some other experimental data from the literature to see if these corrected coefficients work beyond your experimental study.

In Conclusions: Reporting the test results in conclusion could be helpful for readers. Currently, most part of the conclusion is that current models are not good. This is not surprising since these models were developed for concrete without adding SCMs. When the authors introduce a correction term to take into account the effects that SCMs have on the mechanical and shrinkage properties, they can discuss their findings in the conclusion part.

Reviewer 4 Report

In the submitted paper, the effects of flay ash and/or blast furnace slag on the properties of concrete are studied. Different cement replacement levels and mixtures were used and the mechanical characterization of the resulting materials is reported. Standardized prediction models were also utilized to discuss the results.

The research described in this paper is interesting and thorough. I would suggest only minor changes before the publication.

  • Add information and results about the workability of the studied concrete mixtures.
  • Detail the acronym in the title.
  • Line 220: do you mean Table 3?
  • Figure 10 is not cited in the text; please, revise.
  • Carefully check the text; there are some typos and unclear sentences (e.g., lines 180-181).

Round 2

Reviewer 1 Report

This paper can be accepted

Reviewer 3 Report

The authors have satisfactorily addressed all the reviewer's comments. Proposing new terms in the models and adding Figure 14 significantly improved the quality of the manuscript.

Also, it is suggested that the authors follow the following format for Ref [1]:

Ramezani, Mahyar, "Design and predicting performance of carbon nanotube reinforced cementitious materials: mechanical properties and dispersion characteristics." (2019). Electronic Theses and Dissertations. Paper 3255.
https://doi.org/10.18297/etd/3255